# Mapping QTL for Phenological and Grain-Related Traits in a Mapping Population Derived from High-Zinc-Biofortified Wheat

**DOI:** 10.3390/plants12010220

**Published:** 2023-01-03

**Authors:** Nagenahalli Dharmegowda Rathan, Gopalareddy Krishnappa, Anju-Mahendru Singh, Velu Govindan

**Affiliations:** 1Indian Agricultural Research Institute (IARI), New Delhi 110012, India; 2ICAR-Indian Institute of Wheat and Barley Research, Karnal 132001, India; 3International Maize and Wheat Improvement Center (CIMMYT), Texcoco 56237, Mexico

**Keywords:** wheat, QTLs, DArTseq, candidate genes, mapping, yield component traits, biofortification

## Abstract

Genomic regions governing days to heading (DH), days to maturity (DM), plant height (PH), thousand-kernel weight (TKW), and test weight (TW) were investigated in a set of 190 RILs derived from a cross between a widely cultivated wheat-variety, Kachu (DPW-621-50), and a high-zinc variety, Zinc-Shakti. The RIL population was genotyped using 909 DArTseq markers and phenotyped in three environments. The constructed genetic map had a total genetic length of 4665 cM, with an average marker density of 5.13 cM. A total of thirty-seven novel quantitative trait loci (QTL), including twelve for PH, six for DH, five for DM, eight for TKW and six for TW were identified. A set of 20 stable QTLs associated with the expression of DH, DM, PH, TKW, and TW were identified in two or more environments. Three novel pleiotropic genomic-regions harboring co-localized QTLs governing two or more traits were also identified. In silico analysis revealed that the DArTseq markers were located on important putative candidate genes such as *MLO-like protein*, *Phytochrome*, *Zinc finger* and *RING-type*, *Cytochrome P450* and *pentatricopeptide repeat*, involved in the regulation of pollen maturity, the photoperiodic modulation of flowering-time, abiotic-stress tolerance, grain-filling duration, thousand-kernel weight, seed morphology, and plant growth and development. The identified novel QTLs, particularly stable and co-localized QTLs, will be validated to estimate their effects in different genetic backgrounds for subsequent use in marker-assisted selection (MAS).

## 1. Introduction

Micronutrient deficiencies affect two billion people globally. Annually, they accounts for 45% of all child death under the age of five [1]. Furthermore, 52 million children are emaciated and 155 million children are stunted due to micronutrient deficiencies [2]. The iron (Fe) and zinc (Zn) deficiencies are more pronounced and affect one-third of the population in developing nations, particularly children and pregnant women [3,4]. Various interventions have been proposed to combat micronutrient malnutrition, namely, pharmaceutical supplementation, industrial fortification, dietary diversification and biofortification [5,6]. Among them, genetic biofortification employing conventional, molecular or transgenic methods to produce nutrient-rich crop varieties is considered as a sustainable, cost-effective long-term strategy for addressing nutritional needs [7]. With this concern, various biofortified wheat varieties were developed and deployed for production in India, such as ‘WB02′, ‘PBW01Zn′ HUW711 and ‘Zinc Shakti,’ which have a 20–40% higher zinc content than local controls [8].

The adaptation of biofortified cultivars by farmers depends mainly on their yield potential. Grain yield is a complex trait and is an outcome of the combined effect of several yield-contributing agro-morphological and physiological traits [9,10]. Important yield component traits such as DH and DM play a key role in determining the crop’s adaptation to different eco-geographical regions, and in turn yield stability [11,12]. These traits are governed by polygenes, and are influenced by environmental parameters such as temperature and day length. In wheat, vernalization (Vrn), photoperiod (Ppd), and earliness *per se* (Eps) genes have a tremendous effect on these traits [13,14]. Similarly, PH has a strong association with several yield-determining traits such as spikelet number per spike (SNPS), and TKW, therefore playing a major role in enhancing the yield potential [15]. In addition, optimum PH is an important candidate in the ideotype concept proposed by Donald [16] for developing high-yielding wheat cultivars. Grain-related traits such as TKW and TW are important traits, which affect both grain yield and quality; TKW has no nutritional value per se, although it has a dilution effect on protein and micronutrients. Therefore, TKW is one of the important breeding objectives, due to its dual effects on yield and quality. In addition, yield and component traits are genetically complex and are highly influenced by micro- and macro-environmental factors, which make their improvement even more difficult with conventional breeding-approaches. The integration of modern plant-breeding tools such as marker-assisted selection (MAS), marker-assisted recurrent selection (MARS), genomic selection (GS), speed breeding (SB), and genome editing (GE) into conventional breeding-methods is important for accelerating the genetic gain in wheat [17]. Molecular breeding is a potential complementary strategy to conventional breeding to improve complex traits such as yield; however, a better understanding of the genetic architecture is important for the effective utilization of molecular tools. Therefore, genetic dissection of yield-contributing traits is essential for the improvement of wheat yield.

Genome-wide association studies (GWAS) and quantitative-trait-loci (QTL) mapping are the two widely used methods to dissect the genetic basis of complex quantitative traits in crop plants. In previous studies, GWAS panels have been phenotyped in a range of production conditions including drought, irrigated, heat, salt stress, and different nitrogen regimes, to identify QTLs associated with grain yield and its contributing traits through GWAS [10,18,19,20,21,22,23,24,25]. Similarly, several QTLs associated with yield and contributing traits have been identified through bi-parental populations-based QTL mapping under diverse production conditions (irrigated, drought, heat, organic) and different genetic backgrounds (synthetic wheat, Rye selections or translocations, non-adapted background) [26,27,28,29,30,31,32,33,34].

The type of genetic materials includingrecombinant inbred lines, back-cross progenies, doubled haploids, F2, etc. and the size of a mapping population, along with the frequency and distribution of molecular markers on the framework linkage-map are important determinants of QTL mapping. The dilution and concentration effects and the confounding effect of Ppd, Vrn and Rht genes on grain micronutrients are the bottlenecks for wheat biofortification [35,36,37,38]. Thus, a genetic understanding of TKW, TW and DH, DM, and PH is essential in order to develop high-yielding adoptable biofortified-cultivars without compromising grain quality and yield. Thus, we made efforts to dissect the genetic mechanisms of DH, DM, PH, TKW and TW in wheat, using 190 RILs derived from high-Zn-biofortified wheat “Zinc Shakti” and a high-density linkage map developed using DArTSeq makers. Furthermore, we will conduct a robust nested-association-mapping (NAM) study, using a set of RIL populations derived from a common wheat variety ‘Kachu’, crossed with few high-Zn parents, including Zinc-Shakti. This will provide a better understanding of haplotype blocks favorable for agronomic- and nutritional-quality-traits improvement.

## 2. Materials and Methods

### 2.1. Plant Material and Field Experiments

A set of 190 recombinant inbred lines (RILs) in F6 generation along with parental lines viz., Kachu (KAUZ//ALTAR-84/(AOS)AWNED-ONAS/3/MILAN/KAUZ/4/HUITES) and Zinc-Shakti (CROC1_/AE.SQUARROSA(210) //INQALAB-91*2/KUKUNA/3/PBW-343*2/KUKUNA) were grown at Norman E. Borlaug Research Station, Ciudad Obregon, Sonora, Mexico. The mapping population was phenotyped for PH and TKW during 2017–2018 (Y1), 2018–2019 (Y2), 2019–2020 (Y3) and DH, DM, and TW during the 2017–2018 (Y1), and 2018–2019 (Y2) crop seasons.

### 2.2. Phenotyping and Data Analysis

All the genotypes of a mapping population were phenotyped for three agro-morphological (DM, DH, PH) and two grain-related traits (TKW and TW). The RILs, along with the parents, were grown in a randomized-complete-block design with two replications, under the condition of assured irrigation. The data on DH was measured by counting the number of days from germination to 50% of the plants heading in a plot. DM was measured by counting the number of days from germination to physiological maturity, when more than 50% of spikes were ripe and had turned yellow. PH was measured from the ground to the tip of the spike excluding awns, at the late grain-filling stage. TKW was measured with the Seed Count digital imaging system (model SC5000; Next Instruments Pty Ltd., Condell Park, NSW, Australia), which was standardized to measure TKW. The Seed Count system can rapidly and accurately measure wheat grain samples, determining the grain number and the grain physical characteristics based on flatbed scanner technology.

The Meta-R (Multi Environment Trial Analysis with R) version 6.0 software was used for phenotypic analysis (https://excellenceinbreeding.org/toolbox/tools/multi-environment-trail-analysis-r-meta-r (accessed on 20 March 2022)). Best linear unbiased predictors (BLUPs) of each RIL were obtained for an individual year and across years, and used in QTL analysis. The broad-sense heritability (h^2^_BS_), pairwise correlation co-efficient (r_g_) between traits, and the coefficient of variance (CV) were also estimated using Meta-R. The statistical equations used are presented below.
BLUPs (individual environment): *Y_ik_*= *µ* + Rep*i*+ Gen*_k_*+ *ε_ik_*

BLUPs (across environments): *Y_ijkl_ = µ +* Env_i_
*+* Rep*j (*Env_i_*) +* Gen_l_
*+* Env_i_ × Gen_l_
*+ ε_ijkl_*

$${h^{2}}_{\mathrm{BS}}\;(\mathrm{individual}\;\mathrm{environment})\;=\;\frac{\sigma 2g}{\sigma 2g+\frac{\sigma 2ge}{\mathrm{nEnvs}}+\frac{\sigma 2e}{\mathrm{nEnvs}+\mathrm{nreps}}}$$
$${h^{2}}_{\mathrm{BS}}(\mathrm{across}\;\mathrm{environments})\;=\;\frac{\sigma 2g}{\sigma 2g+\frac{\sigma 2ge}{\mathrm{nEnvs}}+\frac{\sigma 2e}{\mathrm{nEnvs}+\mathrm{nreps}}}\;$$
$$r_{g}=\frac{\sigma g\;\left(ij\right)}{\sigma g\left(i\right)\sigma g\left(j\right)}\;$$
$$\mathrm{CV}\;\left(\%\right)\;=\;\;\frac{\mathrm{ASED}}{\;Grand\;mean}\times 100$$
where, *Y_ik_*—trait of interest, *µ*—mean effect, Rep*i*—effect of the *i*th replicate, Gen*_k_*—effect of the *k*th genotype, *ε_ijk_*—error associated with the *i*th replication and the *k*th genotype, Env_i_—effects of the ith environment, Env_i_ × Gen_l_—environment by genotype interaction, *σ*^2^_g_—genotype variance, *σ*^2^_e_—error variance, nreps—number of replicates, *σ*^2^_ge_—genotype-by-environment-interaction variance component, nEnvs—number of environments in the analysis, and ASED—average standard error of the differences between pairs of means.

### 2.3. Genotyping, Linkage Map Construction, and QTL Analysis

The RILs were genotyped with 40,059 DArTseq markers [39]. The DArTseq markers were generated by following genotyping-by-sequencing (GBS) techniques, which employs a combination of restriction enzymes and reduce the genome complexity, to obtain a representation of the whole genome. The quality filtering of the FASTQ files was carried out using a Phred quality score of 30. More stringent filtering was also performed on the barcode sequences, using a Phred quality score of 10, which represents 99.9% of base-call accuracy for at least 75% of the bases. Allele calls for SNP were generated using a proprietary analytical pipeline developed by DArT P/L. The markers were quality filtered using the bin function in the QTL IciMappingv4.2 software (http://www.isbreeding.net) (accessed on 1 November 2022), to remove parental non-polymorphic markers, markers with >30% missing data, and the redundant markers. Both linkage-map construction and QTL mapping were carried out with the QTL IciMappingv4.2 software. The linkage map was constructed with the LOD threshold of 3.0 between adjacent markers [40], the markers were ordered with the nnTwoOpt algorithm and rippled using 5 cM window size. The QTLs were mapped by applying the LOD threshold of 2.5, following the inclusive-composite-interval-mapping (ICIM) method.

### 2.4. In Silico Analysis of QTL and QTL Additive-Effects Estimation

The in silico analysis was carried out in the Ensemble Plants database (http://plants.ensembl.org/index.html (accessed on 1 November 2022)) of the bread-wheat genome with the basic local alignment search tool (BLAST). The sequence information of the markers falling within the confidence interval of the QTL was used to identify putative candidate genes within the vicinity of the markers.

The RILs were grouped based on genotypic classes of multiple stable and major-effect QTLs, using flanking markers, and they were compared with the actual phenotype traits across environments, to identify the QTL additive-effect and best combinations of QTLs for each trait.

## 3. Results

### 3.1. Genetic Variability and Correlation Analysis

Individual environments and the pooled mean across environments, including phenotype ranges of RILs and their parents are presented as box plots in Figure 1. The lowest mean was recorded in Y1 for all three agro-morphological traits, whereas the highest mean was observed for both grain-related traits in Y1. The frequency distribution of the population is presented in Appendix A. A wide range of variation was observed among the RILs, and few transgressive segregants surpassing both the parents appeared for all the traits. The population exhibited a continuous and near-normal distribution for all the traits, suggesting a polygenic nature. The h^2^_BS,_ CV and genotypic variance are provided in Table 1. The h^2^_BS_ range was 0.83–0.95 for TKW, 0.19–0.81 for TW, 0.92–0.96 for DH, 0.83–0.91 for DM and 0.84–0.99 for PH. The genetic variance was highly significant (*p* < 0.0001) for all traits in all environments, except for TW in Y2.

The correlation of agro-morphological and grain-related traits is presented in Table 2. The correlation among DH, DM, and PH was highly significant and positive (*p* < 0.001). However, the correlation between TKW and all three agro-morphological traits was significant and highly negative except in Y3 for PH (*p* < 0.01–0.001). Furthermore, the correlation between TW and DH, and TW and DM were highly significant and negative in Y2 and for the pooled mean (*p* < 0.05–0.001). The correlation between TW and PH was significant and positive for the pooled mean (*p* < 0.01). The correlation between grain-related traits was significant and negative in Y2 (*p* < 0.01). We also observed no correlation or negative correlation for GZnC and GFeC with TW, DH, DM (*p* < 0.05–0.001), but positive correlation with TKW (*p* < 0.001). Interestingly, PH showed positive correlation with GFeC (*p* < 0.05–0.001) and exhibited no association with GZnC [32].

### 3.2. Linkage-Map Construction

The linkage map was constructed using 909 DArTseq markers. The parental non-polymorphic markers, markers with <0.05 and >0.95 allele frequency, >30% missing data and >20% heterozygosity were discarded, to obtain high-quality informative markers. The linkage map spanned a genetic distance of 4665 cM, with an average marker density of 5.13 cM. The subgenome and chromosome-wise distribution of markers are given in Table 3. The highest number of markers were mapped on subgenome A (412) followed by subgenome B (370), and the lowest number of markers (127) were mapped on subgenome D. A detailed description of the framework linkage-map is provided in [32].

### 3.3. QTL Mapping

#### 3.3.1. Agro-Morphological Traits

A total of 23 QTLs including 6 for DH, 5 for DM, and 12 for PH were identified and presented in Table 4 and Figure 2. For DH, the six QTLs were detected on chromosomes 2B, 5A, 5B, and 7D, with the phenotypic variance explained (PVE) ranging from 3.20 to 37.98%. All the identified QTLs were stable, as they were identified in two environments (Y1 and Y2) along with the pooled mean, except for *QDH-5A.3*, which was identified in Y2 and the pooled mean. Two major QTLs, i.e., *QDH-5A.1*, located at 122 cM on 5A, and *QDH-7D*, located at 104 cm on 7D, recorded the highest PVE, ranging from 26.21 to 37.98% and 12.52 to 16.49%, respectively. All the identified QTLs had positive alleles from the Kachu parent except *QDH-7D*, which had positive alleles from the Zinc-Shakti parent.

For DM, five QTLs were mapped on chromosomes 2B, 5A, 5B, 6A, and 7D, with a PVE ranging from 3.33 to 38.23%. Three (QDM-2B, QDM-5A, and QDM-7D) out of five QTLs were consistently expressed in two environments, along with the pooled mean. A major and consistent QTL, i.e., *QDM-5A*, flanked between 5411867 and 1141498 was identified on chromosome 5A at 122 cM, with PVE ranging from 31.54 to 38.23%. Similarly, another major and consistent QTL, i.e., *QDM-7D*, flanked between 2249010 and 100024878 was mapped on chromosome 7D at 105–106 cM, with a PVE ranging from 11.65 to 11.81%. However, the third consistent QTL, i.e., *QDM-2B*, flanked between 3570063 and 3935335 on chromosome 2B and located at 289–290 cM was recorded with a low PVE, ranging from 3.39 to 6.29%. The remaining two QTLs, i.e., *QDM-5B* and *QDM-6A* were identified in one environment each on chromosomes 5B and 6A, with a PVE of 4.32% and 3.33%, respectively. All the QTLs of DM except QTL on chromosome 7D had positive alleles from Kachu, whereas *QDM-7D* had positive alleles from Zinc-Shakti.

The highest number of QTLs (12) were identified for PH on chromosomes 1B, 2B, 3A, 3B, 3D, 4A, 5A, 7D, 6A, 7B, and 7D. The most stable QTL, i.e., *QPH-3D*, was identified in all the tested environments (Y1, Y2, and Y3) along with the pooled mean, which was mapped on chromosome 3D at 135–136 cM, and flanked between 985805 and 5411730, with a PVE ranging from 6.70 to 11.54%. Another three consistent QTLs (*QPH-2B.1, QPH-3A, QPH-5A*) were identified in two environments (Y1 and Y2), along with the pooled mean. Among the stable QTLs, *QPH-5A* recorded the highest PVE, ranging from 13.92 to 25.73%, followed by *QPH-3A* (PVE: 5.92–11.99%), *QPH-3D* (PVE: 6.70–11.54%), and *QPH-2B.1* (PVE: 4.05–8.35%). One QTL, i.e., *QPH-7B.1*, flanked between 100016361 and 3946798 on chromosome 7B located at 187 cM was identified in the Y2 environment and pooled mean, with a PVE ranging from 4.55 to 6.0%. The remaining seven QTLs, i.e., *QPH-7B.2, QPH-1B, QPH-2B.2, QPH-3B, QPH-4A, QPH-5D*, and *QPH-7D* were identified in one environment each, and the maximum number of five QTLs were identified in the Y3 environment, whereas, *QPH-1B* was identified in Y2 and *QPH-7D* identified in Y1. All the identified QTLs had positive alleles from the Zinc-Shakti parent, except *QPH-5A* and *QPH-3B*, which had positive alleles from the Kachu parent.

#### 3.3.2. Grain-Related Traits

A total of 14 QTLs, including eight for TKW and six for TW were identified and presented in Table 4 and Figure 2. The identified QTLs for TKW were mapped on chromosomes 3A, 4A, 5A, 5B, 6A, and 7D, with a PVE ranging from 4.93 to 32.76 %. Out of eight QTLs, four, i.e., *QTKW-3A.1* (Y1 and across yrs), *QTKW-5B* (Y1 and Y2)*, QTKW-6A.1* (Y2 and Y3) and *QTKW-7D* (Y1 and Y3) expressed stably in two environments, with a PVE ranging from 4.93 to 32.76%, 5.98 to 14.69%, 10.1 to 17.4%, and 5.6 10.31%, respectively. The remaining four QTLs (*QTKW-3A.2*, *QTKW-4A*, *QTKW-5A,* and *QTKW-6A.2*) were identified in one environment each. Four QTLs, i.e., *QTKW-3A.1, QTKW-7D, QTKW-3A.2*, and *QTKW-6A.2* had positive alleles from the Kachu parent, and the remaining four QTLs (*QTKW-5B, QTKW-6A.1, QTKW-4A,* and *QTKW-5A*) had positive alleles from the Zinc-Shakti parent.

For TW, a total of six QTLs (*QTW-1B, QTW-6A.1, QTW-6A.2, QTW-6A.3, QTW-1A, QTW-7D*) were identified on chromosomes 1A, 1B, 6A and 7D with a PVE ranging from 3.66 to 42.31%. All the six QTLs were identified in one environment each, except for two QTLs (*QTW-1B* and *QTW-6A.1*)*,* which were identified in the pooled mean as well. The major QTL, i.e., *QTW-7D*, flanked between 3533158 and 1276810, explained the highest PVE, 42.31% in Y1, which was mapped on 7D at 180 cM. The three QTLs (*QTW-1B, QTW-1A*, and *QTW-7D*) had positive alleles from the Kachu parent, whereas the other three QTLs (*QTW-6A.1, QTW-6A.2,* and *QTW-6A.3*) had positive alleles from Zinc-Shakti parent.

#### 3.3.3. Co-localized QTLs

A common genomic region flanked between 5411867 and 100024127 harbors QTLs (*QPH-5A, QDH-5A.1,* and *QDM-5A*) for all the three agro-morphological traits (PH, DH, and DM). These three co-localized QTLs mapped on chromosome 5A at 122 cM. Additionally, these co-localized QTLs, i.e., *QPH-5A, QDH-5A.1*, and *QDM-5A* were also stable, as they were identified in two environments (Y1 and Y2) along with the pooled mean, with a PVE ranging from 13.92 to 25.73%, 26.21 to 37.98%, and 31.54 to 38.23%, respectively. Similarly, another common genomic region flanked between 1206846 and 1082014 harbors QTLs for TKW (*QTKW-6A.2*) and TW (*QTW-6A.3*) on chromosome 6A at 108 cM, with a PVE of 14.63% and 6.75%, respectively. The third common genomic region, flanked between 1698406 and 100027274 was associated with both DM (*QDM-6A*) and TKW (*QTKW-6A.1*) on chromosome 6A at 111 cM, with a PVE of 3.33% and 10.1 to 17.4%, respectively. Additionally, QTKW-6A.1 was stable, as it was identified in two environments (Y2 and Y3).

Additionally, the genomic regions governing GFeC are pleiotropic, with DM, TKW, and TW on chromosome 6A at 110.5–112.5 cM, 107.5–108.5 cM, and 106.5–112.5 cM, respectively. Similarly, GZnC QTLs cluster with DH and DM on chromosome 7D at 105.5–111.5 cM, with PH on chromosome 2B at 243.5–255.5 cM, and with TKW on chromosome 5A, between 125.5 and 127.5 cM (reported in [32]).

#### 3.3.4. Quantitative-Trait-Locus Additive-Effects

The additive-effects of the stable and pleiotropic QTLs were investigated for agro-morphological and grain-related traits (Table 5). The combination of two or more QTLs has an additive-effect for agro-morphological traits (PH, DM, and DH); however, there is no significant additive-effect for grain-related traits (TKW and TW). Three QTL (*QDH-5A.1, QDH-5A.2,* and *QDH-7D*) combinations had the highest average across the environments for DH, and this combination was found in six RILs. For DM, the highest average was observed in the QTL combination of *QDM-2B, QDM-5A,* and *QDM-7D,* and this combination was identified in seven RILs. The three QTL combinations including *QPH-3A, QPH-3D*, and *QPH-5A* yielded the highest average for PH.

#### 3.3.5. Putative Candidate Genes Associated with QTLs

The significant flanking-markers associated with DH, DM, PH, TKW, and TW were used to identify the putative candidate genes using the annotated wheat reference-sequence (RefSeq V1.0), and are presented in Table 6. The functional role of some of the important putative candidate genes associated with stable and pleiotropic QTLs for agro-morphological and grain-related traits was discussed. Two QTLs, *QDH-2B* and *QDM-2B,* encode the *MLO-like protein* (TraesCS2B02G097900), three QTLs (*QDH-5A.1, QDM-5A* and *QPH-5A*) are associated with days to heading and maturity, and plant height is encoded in *Phytochrome* (TraesCS5A02G391300). Similarly, one QTL, i.e., *QPH-3D,* encodes *Zinc finger*, and *RING-type* (TraesCS3D02G095600) also encodes *Cytochrome P450* (TraesCS3D02G077400). The grain-weight QTL, *QTKW-3A.1,* encodes *pentatricopeptide repeat* (TraesCS3A02G232800).

## 4. Discussion

The essential micronutrients play a pivotal role in the proper functioning of metabolic pathways and physiological activities of both plants and humans. The micronutrient deficiency hinders both physical and mental development, and causes various health-related complications such as anemia, infection, infant mortality, diarrhea, stunting, wasting, etc. Efforts have been made to develop biofortified cultivars to address the micronutrient deficiency, and some have been released for commercial cultivation in India, namely, WB-02, PBW01Zn, HUW711 and Zinc-Shakti. However, the adaptation of these varieties by farmers hugely depends on their yield potential. Hence, we tried the genetic dissection of yield-contributing phenological and grain-related traits, using a population developed with high-zinc-biofortified wheat, “Zinc Shakti” and the widely adopted cultivar “Kachu”. The phenotype-based selection in conventional wheat-breeding has improved wheat yield for several decades across the major wheat-growing regions; however, yield plateaus have been observed in many crops including wheat, and the genotype-based modern breeding tools may complement the conventional breeding approaches to break the yield plateaus. Recent efforts have led to the sequencing of the complex wheat genome, which could be valuable in the varietal improvement programs through the identification of marker systems, putative candidate genes, etc. Several QTLs/MTAs have been identified for yield and contributing traits; however, additional genomic studies using varied genetic materials may be useful, as the saturation point has not been reached [41]. Furthermore, phenotyping plays a critical role in the identification of QTLs, as many of the identified QTLs may be genetic-material-specific or environment-specific QTLs. Additionally, the wheat genome is highly complex, and there is always the possibility of detecting novel genomic regions for yield and component traits.

### 4.1. Mapping Population

The RILs were developed by crossing the high-Zn parent ‘Zn-Shakti’, derived from a synthetic hexaploid wheat (Pedigree: CROC1_/ AE.SQUARROSA(210) // INQALAB91*2 / KUKUNA /3 / PBW343*2 / KUKUNA) and the high yielding and highly adopted line in South Asia, ‘Kachu’, popularly known as ‘DPW-621-50′ (Pedigree: KAUZ//ALTAR-84/(AOS)AWNED-ONAS/3/MILAN/KAUZ/4/HUITES). The parents were distantly related (coefficient of parentage (COP = 0.17) and SNP-based diversity (π = 0.26)). This is highly likely to be the reason for obtaining stable QTLs for both nutritional [32] and yield-related traits in our study. The significant effect of environment and genotype-environment interactions (GEI) was observed in the expression of yield and component traits (Figure 1). The RIL population has been extensively tested consecutively for three years, as the magnitude of GEI is an important factor in the detection of environment-specific QTL(s) as well as consistent QTL(s).

### 4.2. Correlation Studies

The highly positive and significant correlation among DH, DM, and PH suggests that some common genetic mechanism existed in the expression of these traits. The strong positive association of agro-morphological traits was further supported by the identification of a common genomic region flanked between 5411867 and 100024127, which harbors QTLs (*QPH-5A*, *QDH-5A.1*, and *QDM-5A*) for the same three agro-morphological traits (PH, DH, and DM). However, the correlation between TKW and all the three agro-morphological traits was significant and highly negative, except in Y3 for PH. Significant correlations found in this study have also been reported in earlier studies [23,42]. The positively associated traits (DH, DM, and PH) can be improved simultaneously; however, for the improvement of negatively associated traits, the adoption of suitable breeding-methods to break the undesirable linkages is envisaged. In our previous studies [32,38], we observed no correlation or a negative correlation for GZnC and GFeC with TW, DH, and DM, but a positive correlation with TKW. PH had a positive correlation with GFeC, and exhibited no association with GZnC. Interestingly, we found co-localized QTLs for GFeC with DM, TKW and TW, and for GZnC with DH, DM, PH, and TKW.

### 4.3. Linkage Map

The linkage map was constructed with 909 DArTseq high-quality informative markers in a set of 190 RILs spanning a map length of 4665 cM, with an average marker density of 5.13 cM, and was utilized for the QTL analysis. A total of thirty-seven QTLs, including twelve for PH, six for DH, five for DM, eight for TKW, and six for TW were identified. The highest number of QTLs were identified for subgenome A (18 QTLs), followed by subgenome B (12 QTLs) and subgenome D (7 QTLs). The subgenome-wise distribution of QTLs is similar to the marker distribution pattern of subgenomes, as the highest number of markers were mapped on subgenome A (412), followed by subgenome B (370), and the lowest number of markers (127) were mapped on subgenome D. The type and size of a mapping population and frequency and distribution of markers on the linkage map are considered to be key determinants for QTL identification. In previous reports also, the pattern of QTL distribution for grain-quality traits was similar [43]; however, the enrichment of subgenome D with additional markers substantially increased the power of QTL identification [44]. Furthermore, a total of fifteen stable and three pleiotropic QTLs were identified.

### 4.4. QTL Analysis

A total of six QTLs were identified for DH on the 2B, 5A, 5B, and 7D chromosomes. Previous reports also identified some QTLs on the same chromosomes, 2B [45,46,47], 5A [19,48,49], 5B [10,19,20,33,47,50], and 7D [30,31,45,46,49,50,51], at different positions. Five QTLs were identified for DM on the 2B, 5A, 5B, 6A, and 7D chromosomes. Similarly, previous reports also identified some QTLs at different positions on the same chromosomes: 2B [45], 5A [51,52], 6A [19,51,52], and 7D [52].

The highest number of 12 QTLs were identified for PH on the 1B, 2B, 3A, 3B, 3D, 4A, 5A, 5D, 7B, and 7D chromosomes. Plant height is one of the important yield component traits in wheat, and therefore many QTLs were also identified in previous reports on chromosomes 1B [10,19,37,40,51], 2B [10,40,45,51,53], 3A [26,54], 3B [10,19,22,40], 4A [26,45,52,53], 5A [19,26,45,51,53,55,56], and 7B [10], at different positions. In addition, Alemu et al. [52] identified a QTL on the 4A chromosome at 41cM, which was similar to that of *QPH-4A* and mapped at 41cM, explaining a 4.58% phenotypic variation. Similarly, Liu et al. [40] identified a QTL at 197cM on the 3B chromosome which was similar to the QTL (*QPH-3B*) identified in the present study at 195 cM on the same chromosome with 7.71% phenotypic variation.

Eight QTLs were identified for TKW on the 3A, 4A, 5A, 5B, 6A, and 7D chromosomes. Previous reports also identified some QTLs on the same chromosomes: 3A [10,19,24,50,56], 4A [19,31,46,50], 5A [23,29,40,57], 5B [20,29,40,45,52,53,54,57], 6A [23,29,31,37,46,48,50,52,53,55,56,57], and 7D [29,31,40,50], at different positions. In addition, three previous reports, i.e., Gahlaut et al. [19], Edae et al. [46], and Ramya et al. [27] identified a QTL at 73 cM, 72.4 cM, and 70 cM, respectively, on the 5B chromosome, which was similar to the QTL (*QTKW-5B*) identified in the present study at 70.5–73.5 cM, on the same chromosome. For TW, six QTLs were identified on the 1A, 1B, 6A, and 7D chromosomes. Some QTLs were also identified in previous reports on the 1A [30,57], 6A [55,58], and 7D [29,30,46] chromosomes, at different positions.

The highest number of five stable QTLs (*QDH-2B*, *QDH-5A.1*, *QDH-5A.2*, *QDH-5B*, and *QDH-7D*) were identified for DH, followed by four stable QTLs (*QPH-3D*, *QPH-2B.1*, *QPH-3A*, and *QPH-5A*) for PH, three stable QTLs for DM (*QDM-2B*, *QDM-5A*, and *QDM-7D)* and three stable QTLs for TKW (*QTKW-5B*, *QTKW-6A.1*, and *QTKW-7D*). However, there was no stable QTL identified for TW. Furthermore, nineteen QTLs, including three for DH (QDH-5A.1, QDH-5A.2, QDH-7D), two for DM (*QDM-5A* and *QDM-7D*), four for PH (*QPH-3D*, QPH-3A, QPH-5A, QPH-5D), six for TKW (*QTKW-3A.1*, *QTKW-5B*, *QTKW-6A.1*, *QTKW-7D*, *QTKW-4A*, and *QTKW-6A.2*), and four for TW (*QTW-1B*, *QTW-6A.1*, *QTW-1A*, and *QTW-7D*) showed more than 10.0% PVE in at least one environment. In addition, two major QTLs (*QDH-5A.1* and *QTW-7D*) showed more than 30.0% of PVE. The parent Kachu contributed more favorable alleles to the QTLs associated with DH and DM, whereas the Zinc-Shakti parent contributed more favorable alleles to the QTLs associated with PH. This is expected, because more selection pressure for DH and DM has been exerted in the widely adapted Kachu parent compared to other parent since the Green Revolution era, and therefore more favorable alleles have been accumulated in the widely adapted Kachu parent. However, the deployment of semi-dwarf genes into the background of tall genotypes across the major wheat-growing regions drastically reduced the height to the optimum level during the Green Revolution period, and afterward there may not have been much selection pressure for PH in the widely adapted Kachu parent, compared to Zinc Shakti. For grain-related traits, both the parents of the mapping population contributed equally favorable alleles to different QTLs. The grain-related traits including TKW and TW are very important traits, and have undergone selection pressure in both parents, thereby accumulating favorable alleles in both parents and also equally contributing to grain-related QTLs.

### 4.5. Pleiotropic QTL Regions

Three common genomic regions flanked between 5411867 and 100024127 on 5A, 1206846 and 1082014 on 6A, and 1698406 and 100027274 on 6A, harboring co-localized QTLs associated with two or more traits, were identified. The significant associations of traits in the desired direction are always beneficial for the simultaneous improvement of the associated traits. One of the parents of the mapping population is a high-zinc parent (Zinc-Shakti) and the other parent is a high yielding, widely adopted parent (Kachu). Therefore, there is always a possibility of identifying novel QTLs for both grain quality and agro-morphological traits. In our earlier study, several QTLs were reported for grain iron- and zinc-concentration [32]. A total of eight genomic regions identified for various agro-morphological and grain-related traits in the present study correspond to our previous study [32] for grain iron- and zinc-concentration. For example, two QTLs on 7D, i.e., *QDH-7D* and *QDM-7D*, mapped in the confidence interval of 100.5–105.5 cM and 104.5–106.5 cM, respectively, correspond to the confidence interval of 105.5–111.5 cM for the grain zinc-concentration of the QTL (*QZnC-7D.1*). Similarly, *QDM-6A* (110.5–112.5 cM), *QPH-2B.1* (243.5–249.5 cM), *QTKW-6A.1* (110.5–111.5 cM), *QTKW-5A* (125.5–126.5 cM), *QTKW-6A.2* (107.5–108.5 cM), and *QTW-6A.3* (106.5–108.5 cM), respectively, correspond to the confidence interval of 110.5–112.5 cM (*QFeC-6A*), 251.5–255.5 cM (*QZnC-2B.2* and *QZnC-2B.3*), 110.5–112.5 cM (*QFeC-6A*), 126.5–127.5 cM (*QZnC-5A*), 110.5–112.5 cM (*QFeC-6A*), and 110.5–112.5 cM (*QFeC-6A*). Therefore, effective utilization of these stable and co-localized QTLs in varietal-improvement programs through MAS may simultaneously improve the grain yield and grain micronutrient-concentration in wheat.

### 4.6. QTL Utilization Strategy

The exploitation of the QTLs of agro-morphological traits depends on the breeding objective to develop lines suitable for different growing environments. The correlation studies indicated that the best ideotype would be reduced DH and DM, and increased PH, for enhancing grain-micronutrient (GFeC and GZnC) and yield (TKW) traits. Therefore, combining PH QTLs *QPH-3A* and *QPH-3D* from Zinc Shakti and *QPH-5A* from Kachu, and individual QTLs, *QDH-7D* and *QDM-7D* from Zinc Shakti, would be of great value for attaining the objective. Combining QTLs caused only a slight increase in TW, while TKW further stresses the multigenic nature and breeding complexity of these yield-determining traits. The pleiotropic regions found on 6A between 106.5 and 112.5 cM for GFeC, DM, TKW and TW indicate the role of this chromosome in governing multiple complex traits.

### 4.7. Putative Candidate Genes

The putative candidate genes underlying stable and pleiotropic QTLs for agro-morphological and grain-related traits were identified through a BLAST search (Table 6). The QTLs identified on various chromosomes were located in gene coding-regions related to protein-binding factors, photoreceptors, transporters, etc. For example, *QDH-2B* and *QDM-2B* associated with days to heading and maturity encode MLO-like protein (TraesCS2B02G097900) genes, which were found to have a functional role in the regulation of agro-morphological traits in rice. For instance, *OsMLO12*, *OsMLO4*, *OsMLO10*, and *OsMLO8* showed preferential expression in mature pollen, in the root tips, throughout the roots except at the tips, and in the leaves and trinucleate pollen, respectively [59]. Furthermore, in *osdxr* mutants that exhibited defects in the light response, *OsMLO1*, *OsMLO3*, *OsMLO8*, and four calmodulin genes were down-regulated.

The three QTLs (*QDH-5A.1*, *QDM-5A* and *QPH-5A*) associated with days to heading, maturity, and plant height, encode *Phytochrome* (TraesCS5A02G391300). Phytochromes play an important role in light-signaling and the photoperiodic control of flowering-time in plants. A candidate gene, i.e., *HvPHYTOCHROME C*, was identified for the early-maturity 5 locus modulating the circadian clock and photoperiodic flowering in barley [60]. Similarly, the ectopic expression of a phytochrome B gene from Chinese cabbage in Arabidopsis promotes seedling de-etiolation, dwarfing in mature plants, and delayed flowering [61].

Plant height is one of the important agro-morphological traits, and significantly contributed to yield gains during the Green Revolution. Putative candidate genes such as *Zinc finger*, *RING-type*, and *Cytochrome P450* regulate plant growth in many crop plants. One QTL, i.e., *QPH-3D,* associated with plant-height, encodes *Zinc finger*, and *RING-type* (TraesCS3D02G095600), found to have a role in flowering-time modulation, abiotic-stress tolerance and plant growth and development. The characterization of the zinc finger transcription-factor gene (*Cm-BBX24*) suggests the dual role of flowering-time modulation and abiotic-stress tolerance in the chrysanthemum, partly by the influence of GA biosynthesis [62]. The other zinc finger transcription-factor gene, i.e., *TANDEM ZINC-FINGER PLUS3* (*TZP*) controls the expression of growth-promoting transcriptional regulators via the direct association with light-regulated promoter elements in Arabidopsis. Elucidating how such novel protein-complexes coordinate gene expression may help breeders to optimize plant growth and development [63]. The Really Interesting New Gene (RING) finger proteins characterized by the RING domain, which contains around 40–60 residues, are thought to be E3 ubiquitin ligases. The RING-finger proteins play significant roles in plant growth, stress resistance, and signal transduction in crop plants [64]. The RING-type E3-ubiqitin-ligase barley genes (*HvYrg1–2*) control the characteristics of both vegetative organs (leaf width and weight, plant weight and height, flowering-time, grain-filling duration, root-growth) and seed components (thousand-grain weight and seed morphology) in barley [65]. A novel RING finger protein, i.e., *PLANT ARCHITECTURE* and *GRAIN NUMBER 1 (PAGN1*) regulates the number of grains per panicle, the plant height, and the number of tillers in rice [66].

The same QTL, i.e., *QPH-3D*, which encodes *Zinc finger*, RING-*type* also encodes *Cytochrome P450* (TraesCS3D02G077400). The cytochrome P450 gene, i.e., *CsCYP85A1,* is a putative candidate for *super compact-1* (*scp-1*), which controls the plant height in cucumber [67]. Another cytochrome P450 family member, *OsCYP96B4*, reduces plant height in a transcript dosage-dependent manner in rice [68]. Amino-acid transporters play a key role in resource-allocation processes that support plant growth and development; an amino-acid transporter, i.e., *LHT1*, plays a role in plant-growth in rice, and its disruption leads to growth inhibition and low yields [69]. The grain-weight QTL *QTKW-3A.1* encodes *pentatricopeptide repeat* (TraesCS3A02G232800), which affect grain-weight in rice and maize. The novel QTL (*qKW9*) related to kernel-size encodes a pentatricopeptide repeat protein that affects photosynthesis and grain-filling in maize [70]. The peptide transporter OsNPF7.3 enhances nitrogen allocation and increases grain-yield in rice through enhanced tiller numbers, panicles per plant, filled-grain numbers per panicle, and grain nitrogen-content [71].

## 5. Conclusions

The study with 190 RILs has shown that DH, DM, PH, TKW, and TW were quantitatively inherited traits. The strong positive correlation among DH, DM, and PH suggested the possibility of improving these traits simultaneously. Furthermore, biofortification traits such as GFeC and GZnC could also be increased simultaneously, as they exhibited a significant and positive correlation with TKW; this may be due to the elimination of the dilution effect in the Zinc-Shakti parent, through careful selection for high GFeC and GZnC, along with high TKW. A total of thirty-seven QTLs, including twelve for PH, six for DH, five for DM, eight for TKW, and six for TW were identified. A total of 20 stable QTLs were identified in more than one environment and associated with the expression of DH, DM, PH, and TKW. Similarly, three novel pleiotropic genomic-regions harboring co-localized QTLs governing two or more traits were also identified. Three QTLs (*QPH-4A*, *QPH-3B*, *QTKW-5B*) identified in the present study were also identified in previous reports, and therefore could be potential candidates for MAS. Several putative candidate genes encode important molecular functions such as tissue-specific expression, light-signaling and photoperiodic modulation of flowering-time, abiotic-stress tolerance, and plant growth and development, which have a role in controlling both agro-morphological and grain-related traits. Further validation and functional characterization of the candidate genes to elucidate the role of these genes in wheat is envisaged.

## Figures and Tables

**Figure 1 plants-12-00220-f001:**
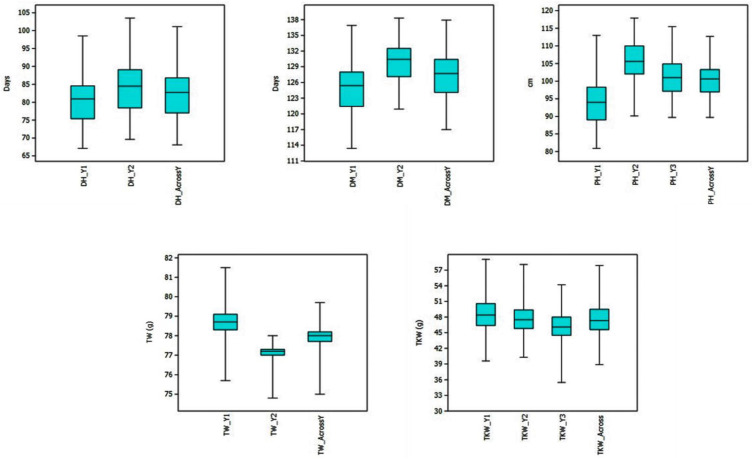
Boxplots indicating the range and mean values of DH, DM, PH, TKW and TW in RIL population from 2017–2018 (Y1), 2018–2019 (Y2), 2019–2020 (Y3) and across years.

**Figure 2 plants-12-00220-f002:**
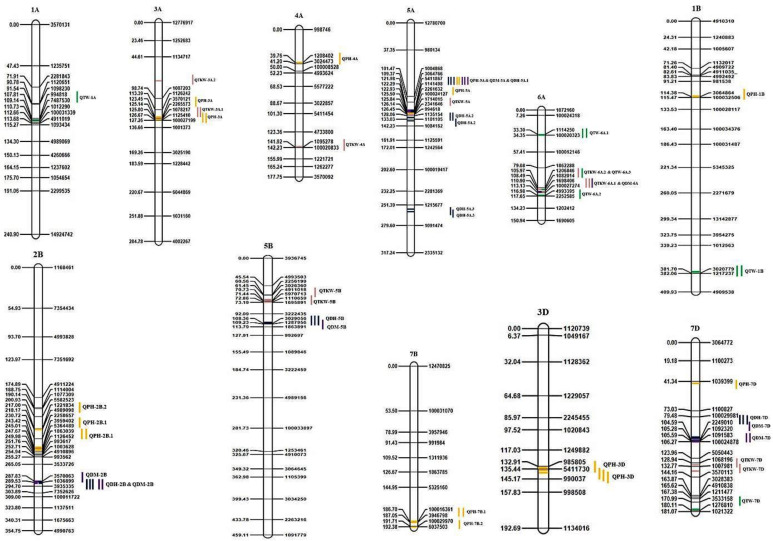
Linkage map and QTLs detected for DH, DM, PH, TKW and TW for 2017–2018 (Y1), 2018–2019 (Y2), 2019–2020 (Y3) and across years. The same QTLs identified in different years are placed based on their signal-detected genomic position, but with same name (for example: QPH-3D, not as QPH-3D.1, 3D.2). QTL color coding: Dark Blue—DH, Purple—DM, Yellow—PH, Pink—TKW, Green—TW.

**Table 1 plants-12-00220-t001:** Estimates of phenotypic values, CV, heritability and variance for the traits in RILs andparents evaluated from 2017–2018 (Y1), 2018–2019 (Y2), 2019–2020 (Y3), and across the years, at Ciudad Obregon, Mexico.

Trait	Env	Kachu	ZincShakti	RIL Population		
		Parent 1	Parent 2	CV (%)	H^2^_BS_	Genotypic Variance
TKW (g)	2017–2018	48.4	47.5	2.32	0.95	11.28 ***
	2018–2019	44.7	45.5	4.02	0.83	8.93 ***
	2019–2020	44.6	43.6	3.4	0.86	7.65 ***
	Across Years	45.9	45.5	3.31	0.95	9.02 ***
TW (g)	2017–2018 *	79.8	76.2	0.76	0.81	0.78 ***
	2018–2019 *	78.2	74.8	3.31	0.19	0.75***
	Across Years *	79	75.5	2.38	0.46	0.75 ***
DH (Days)	2017–2018	81	74	3.24	0.92	40.84 ***
	2018–2019 **	84.5	70.5	3.41	0.93	53.22 ***
	Across Years *	82.8	72.3	3.32	0.96	46.58 ***
DM (Days)	2017–2018 *	127.5	119	2.05	0.89	26.11 ***
	2018–2019	134	131	2.18	0.83	19.30 ***
	Across Years	130.8	125	2.12	0.91	21.81 ***
PH (cm)	2017–2018 *	82.8	89.5	3.52	0.9	51.15 ***
	2018–2019 *	97	103	0.66	0.99	36.57 ***
	2019–2020 **	97.5	107.5	1.14	0.98	31.32 ***
	Across Years **	92.4	100.3	2.06	0.84	26.48 ***

The *t*-test results indicating significant difference between parents for each trait are provided in the environment column. * significant values at *p* < 0.05, ** significant values at *p* < 0.01, *** significant values at *p* < 0.001.

**Table 2 plants-12-00220-t002:** Genetic correlation among DH, DM, PH, TKW and TW for 2017–2018 (Y1), 2018–2019 (Y2), 2019–2020 (Y3) and across the years.

Traits	Env.	DH	DM	PH	TKW
DM	2017–2018	0.93 ***			
	2018–2019	0.90 ***			
	2019–2020	NA			
	Across Years	0.94 ***			
PH	2017–2018	0.61 ***	0.44 ***		
	2018–2019	0.49 ***	0.41 ***		
	2019–2020	NA	NA		
	Across Years	0.77 ***	0.63 ***		
TKW	2017–2018	−0.48 ***	−0.55 ***	−0.31 ***	
	2018–2019	−0.4 2***	−0.55 ***	−0.20 **	
	2019–2020	NA	NA	−0.09	
	Across Years	−0.44 ***	−0.54 ***	−0.27 ***	
TW	2017–2018	−0.14	−0.12	0.13	0.04
	2018–2019	−0.28 ***	−0.36 ***	−0.12	−0.20 **
	2019–2020	NA	NA	NA	NA
	Across Years	−0.18 *	−0.19 **	0.21 **	−0.03

* Significant at *p* < 0.05, ** Significant at *p* < 0.01, *** Significant at *p* < 0.001.

**Table 3 plants-12-00220-t003:** Chromosome and subgenome-wise distribution of markers in the RIL population.

Chromosome	1	2	3	4	5	6	7	Total
Genome	RIL Population							
A	62	80	54	67	45	67	37	412
B	46	109	46	9	90	31	39	370
D	11	17	38	01	05	12	43	127

**Table 4 plants-12-00220-t004:** QTLs identified for DH, DM, PH, TKW and TW for 2017–2018 (Y1), 2018–2019 (Y2), 2019–2020 (Y3) and across years.

QTL Name	Envi	Chr	Position (cM)	Flanking Markers	LOD	PVE(%)	Add	Confidence Interval
Days to Heading (DH)								
*QDH-2B*	1, 2, 4	2B	290	1036899–3935335	5.17, 6.49, 6.10	5.17, 5.67, 5.75	1.40, 1.71, 1.60	288.5–292.5
*QDH-5A.1*	1, 2, 4	5A	122	5411867–1141498	30.31, 24.79, 28.92	37.98, 26.21, 33.82	3.87, 3.75, 3.94	121.5–122.5
*QDH-5A.2*	1, 2, 4	5A	132–134	1135154–1084162	6.39, 11.11, 9.04	6.53, 10.76, 8.98	1.56, 2.33, 1.97	129.5–137.5
*QDH-5B*	1, 2, 4	5B	109	3029056–1287956	4.78, 3.87, 4.33	4.42, 3.20, 3.83	1.30, 1.28, 1.30	106.5–109.5
*QDH-7D*	1, 2, 4	7D	104	100029981–2249010	11.89, 15.86, 13.96	12.52, 16.49, 14.46	−2.17, −2.90, −2.52	100.5–105.5
*QDH-5A.3*	2, 4	5A	257	1215677–1091474	3.53, 2.74	4.30, 3.75	1.47, 1.27	246.5–271.5
Days to Maturity (DM)								
*QDM-2B*	1, 2, 4	2B	289–290	3570063–3935335	4.08, 5.39, 6.80	3.39, 5.69, 6.29	1.02, 1.07, 1.26	287.5–292.5
*QDM-5A*	1, 2, 4	5A	122	5411867–1141498	31.10, 23.41, 30.43	36.65, 31.54, 38.23	3.42, 2.58, 3.16	121.5–122.5
*QDM-7D*	1, 2, 4	7D	105–106	2249010–100024878	12.62, 10.98, 12.02	11.69, 11.81, 11.65	−1.88, −1.54, −1.70	104.5–106.5
*QDM-5B*	1	5B	110	1287956–1863891	4.95	4.32	1.15	108.5–112.5
*QDM-6A*	2	6A	111	1698406–100027274	3.36	3.33	0.81	110.5–112.5
Plant Height (PH)								
*QPH-3D*	1,2,3,4	3D	135–136	985805–5411730	7.25, 6.92, 8.59, 8.84	6.70, 8.4, 11.54, 9.66	−2.04, −1.86, −2.06, −1.61	132.5–140.5
*QPH-2B.1*	1,2,4	2B	245–248	3959402–1126452	6.25, 7.15, 4.14	5.61, 8.35, 4.05	−1.87, −1.85, −1.04	243.5–249.5
*QPH-3A*	1,2, 4	3A	124–127	3570121–100027199	6.54, 9.15, 11.22	5.92, 11.23, 11.99	−1.96, −2.17, −1.82	123.5–127.5
*QPH-5A*	1,2,4	5A	122–123	5411867–100024127	23.38, 11.25, 17.15	25.73, 13.92, 19.20	4.13, 2.5, 2.34	121.5–123.5
*QPH-7B.1*	2,4	7B	187	100016361–3946798	4.04, 6.15	4.55, 6.0	−1.37, −1.27	186.5–188.5
*QPH-7B.2*	3	7B	192	100029970–6037503	4.04	4.54	−1.3	191.5–192
*QPH-1B*	2	1B	115	3064864–100032506	2.67	3.03	−1.11	114.5–116.5
*QPH-2B.2*	3	2B	218	1221834–4989098	5.81	6.7	−1.59	217.5–218.5
*QPH-3B*	3	3B	195	3022574–1088815	6.18	7.71	1.73	194.5–195.5
*QPH-4A*	3	4A	41	1208402–3024473	4.06	4.58	−1.3	39.5–45.5
*QPH-5D*	3	5D	6	100021862–1089439	8.24	11.3	−2.09	2.5–8.5
*QPH-7D*	1	7D	42	1039399–1100827	4.94	4.55	−1.68	39.5–52.5
Thousand-Kernel Weight (TKW)								
*QTKW-3A.1*	1, 4	3A	126	1078217–1125410	2.98, 55.88	4.93, 32.76	0.70, 3.87	125.5–127.5
*QTKW-5B*	1,2	5B	71–73	4911018–1695891	3.49, 37.09	5.98, 14.69	−0.77, −2.34	70.5–73.5
*QTKW-6A.1*	2, 3	6A	111	1698406–100027274	29.48, 23.12	10.1, 17.4	−1.90, −1.69	110.5–111.5
*QTKW-7D*	1, 3	7D	132–137	1068196–3570113	5.54, 7.81	10.31, 5.6	1.04, 0.98	129.5–142.5
*QTKW-3A.2*	3	3A	62	1134717–1007203	3.53	6.25	1.01	44.5–76.5
*QTKW-4A*	1	4A	142	1095278–100020833	7	12.22	−1.09	140.5–142.5
*QTKW-5A*	1	5A	126	1714015–2341646	5.49	9.89	−1	125.5–126.5
*QTKW-6A.2*	2	6A	108	1206846–1082014	37.54	14.63	2.3	107.5–108.5
Test Weight (TW)								
*QTW-1B*	2, 4	1B	382	3020779–1217237	3.31, 4.03	9.30, 10.23	0.10, 0.16	381.5–382.5
*QTW-6A.1*	2, 4	6A	34	1114250–100020323	2.83, 4.01	6.85, 10.23	−0.09, −0.15	33.5–35.5
*QTW-6A.2*	1	6A	117	4993395–2252585	11.58	3.66	−0.3	116.5–117.5
*QTW-6A.3*	4	6A	108	1206846–1082014	2.6	6.75	−0.13	106.5–108.5
*QTW-1A*	4	1A	109	994818–7487530	3.97	10.01	0.15	107.5–109.5
*QTW-7D*	1	7D	180	3533158–1276810	64.74	42.31	1.05	178.5–181

1—2017-18, 2—2018-19, 3—2019-20, 4—Across years, Blue—Pleiotropic with GZnC, Red—Pleiotropic with GFeC.

**Table 5 plants-12-00220-t005:** The Phenotypic effect on combining high-effect and stable QTLs of DH, DM, PH, TKW and TW.

QTL	Flanking Markers	Marker Type	No. of RILs	Environments			
QTL Additive-Effect for Days to Heading (DH)				Y1	Y2	Y3	Across Yrs
5A.1	5411867 + 1141498	A + A	107	84.05	88.30	-	86.31
5A.2	1135154 + 1084162	A + A	74	84.06	88.63	-	86.48
7D	100029981 + 2249010	B + B	16	82.70	87.57	-	85.24
5A.1 + 5A.2	5411867 + 1141498 + 1135154 + 1084162	A + A + A + A	70	84.64	89.22	-	87.09
5A.1 + 7D	5411867 + 1141498 + 100029981 + 2249010	A + A + B + B	11	85.39	89.70	-	87.73
5A.2 + 7D	1135154 + 1084162 + 100029981 + 2249010	A + A + B + B	7	84.64	88.65	-	86.79
5A.1 + 5A.2 + 7D *	5411867 + 1141498 + 1135154 + 1084162 + 100029981 + 2249010	A + A + A + A + B + B	6	86.33	89.58	-	88.15
QTL additive-effect for Days to Maturity (DM)							
2B	3570063 + 3935335	A + A	67	125.25	130.22	-	127.77
5A	5411867 + 1141498	A + A	103	126.90	131.58	-	129.35
7D	2249010 + 100024878	B + B	81	125.13	130.45	-	127.83
2B + 5A	3570063 + 3935335 + 5411867 + 1141498	A + A + A + A	33	127.99	132.47	-	130.39
2B + 7D	3570063 + 3935335 + 2249010 + 100024878	A + A + B + B	21	126.32	131.09	-	128.78
5A + 7D	5411867 + 1141498 + 2249010 + 100024878	A + A + B + B	35	128.50	132.74	-	130.80
2B + 5A + 7D *	3570063 + 3935335 + 5411867 + 1141498 + 2249010 + 100,024878	A + A + A + A + B + B	7	130.77	134.41	-	132.87
QTL additive-effect for Plant Height (PH)							
3A	3570121 + 100027,199	B + B	95	95.56	107.35	102.79	101.71
3D	985805 + 5411730	B + B	47	95.84	107.44	103.55	102.04
5A	5411867 + 100024127	A + A	86	97.14	107.78	102.01	102.10
3A + 3D	3570121 + 100027199 + 985805 + 5411730	B + B + B + B	24	97.73	108.91	105.02	103.46
3D + 5A	985805 + 5411730 + 5411867 + 100024127	B + B + A + A	24	98.68	109.89	103.79	103.67
3A + 5A	3570121 + 100027199 + 5411867 + 100024127	B + B + A + A	43	98.99	109.50	102.98	103.43
3A + 3D + 5A *	3570121 + 100027199 + 985805 + 5411730 + 5411867 + 100024127	B + B + B + B + A + A	10	102.71	112.13	105.68	106.09
QTL additive-effect for Thousand-Kernel Weight (TKW)							
3A.1	1078217 + 1125410	A + A	71	49.23	48.31	46.79	48.17
5B	4911018 + 1695891	B + B	90	48.92	48.35	46.71	48.05
6A.1	1698406 + 100027274	B + B	72	48.31	47.83	46.16	47.44
3A.1 + 5B	1078217 + 1125410 + 4911018 + 1695891	A + A + B + B	39	49.74	48.78	47.07	48.62
3A.1 + 6A.1	1078217 + 1125410 + 1698406 + 100027274	A + A + B + B	32	49.49	48.65	46.90	48.42
5B + 6A.1	4911018 + 1695891 + 1698406 + 100027274	B + B + B + B	43	48.97	48.47	46.85	48.16
3A.1 + 5B + 6A.1	1078217 + 1125410 + 4911018 + 1695891 + 1698406 + 100027274	A + A + B + B + B + B	20	49.59	48.62	47.08	48.51
QTL additive-effect for Test Weight (TW)							
1B	3020779 + 1217237	A + A	61	78.95	77.24	-	78.14
6A.1	1114250 + 100020323	B + B	59	78.92	77.22	-	78.11
7D	3533158 + 1276810	A + A	61	78.75	77.14	-	77.96
1B + 6A.1	3020779 + 1217237 + 1114250 + 100020323	A + A + B + B	24	79.15	77.32	-	78.30
1B + 7D	3020779 + 1217237 + 3533158 + 1276810	A + A + A + A	32	78.74	77.20	-	78.04
6A.1 + 7D	1114250 + 100020323 + 3533158 + 1276810	B + B + A + A	34	78.85	77.24	-	78.11
1B + 6A.1 + 7D	3020779 + 1217237 + 1114250 + 100020323 + 3533158 + 1276810	A + A + B + B + A + A	13	78.97	77.13	-	78.01

* Highest average-effect on combining the QTLs. A—Parent 1 (Kachu) marker-type, B—Parent 2 (Zinc-Shakti) marker-type; no. of RILs indicates the total number of RILs in the population containing particular QTL combination.

**Table 6 plants-12-00220-t006:** Putative candidate genes governing the expression of DH, DM, PH, TKW and TW and their molecular functions.

QTLs	Chr	Physical Position (Mb)	TraesID	Putative Candidate Genes	Molecular Function
QDH-2B QDM-2B	2B	46.67–57.78	TraesCS2B02G097900	MLO-like protein	Calmodulin binding
			TraesCS2B02G083700	UDP-glucuronosyl/UDP-glycosyltransferase	UDP-glycosyltransferase activity
QDH-5A.1, QDM-5A QPH-5A	5A	581.13–586.6	TraesCS5A02G391300	Phytochrome	Photoreceptor activity
			TraesCS5A02G388800	Leucine-rich repeat-domain superfamily, protein kinase-like domain superfamily	Protein serine/threonine kinase activity, protein binding
			TraesCS5A02G388900	Glycoside hydrolase superfamily	Hydrolase activity, hydrolyzing O-glycosyl compounds
			TraesCS5A02G383400	Heavy-metal transporting P1B-ATPase 3	Transporter activity
QDH-7D QDM-7D	7D	58.63–59.47	TraesCS7D02G099300	Gnk2-homologous domain superfamily, Protein kinase-like domain superfamily	Protein kinase activity, ATP binding
			TraesCS7D02G096600	Acetyl-CoA carboxylase, ClpP/crotonase-like domain superfamily	Ligase activity
QDH-5A.2	5A	572.02–584.72	TraesCS5A02G387800	RNA-binding domain superfamily, Nucleotide-binding alpha-beta plait domain superfamily	Nucleic acid binding
			TraesCS5A02G380100	Deoxyhypusine synthase	Peptidyl-lysine modification to peptidyl-hypusine
			TraesCS5A02G374000	9-cis-epoxycarotenoid dioxygenase	Carotenoid dioxygenase activity
QPH-3D	3D	37.96–48.84	TraesCS3D02G095600	Zinc finger, RING-type	Ubiquitin-protein-ligase activity
			TraesCS3D02G077400	Cytochrome P450	Heme binding, oxidoreductase activity
			TraesCS3D02G096500	Amino acid/polyamine transporter I	Transmembrane transporter activity
QPH-3A and QTKW-3A.1	3A	143.53–510.83	TraesCS3A02G282400	Peptidase S8 propeptide/proteinase inhibitor I9 superfamily	-
			TraesCS3A02G261800	Serine-threonine/tyrosine-protein kinase, wall-associated receptor kinase	Protein kinase activity, polysaccharide binding
			TraesCS3A02G232800	Tetratricopeptide-like helical domain superfamily, pentatricopeptide repeat	Protein binding
			TraesCS3A02G153000	Cytochrome P450	Heme binding, oxidoreductase activity
QTKW-6A.1	6A	490.48–561.59	TraesCS6A02G265500	C2 domain superfamily, PH-like domain superfamily, prolycopenC2 and GRAM domain-containing protein, VASt domain	-
			TraesCS6A02G265600	RNA-binding domain superfamily, Nucleotide-binding alpha-beta plait domain superfamily	Nucleic acid binding, RNA binding
			TraesCS6A02G265800	Tetrapeptide transporter, Oligopeptide transporter superfamily	Oligopeptide transmembrane transporter-activity
QTKW-6A.2 and QTW-6A.3	6A	106.47–183.52	TraesCS6A02G134100	Domain of unknown function DUF4220, protein of unknown function DUF594	-
			TraesCS6A02G134000	Sugar/inositol transporter, MFS transporter superfamily	Carbohydrate/monosaccharide transmembrane transporter-activity
			TraesCS6A02G172500	Double-stranded RNA-binding domain, Ribonuclease III	RNA binding
			TraesCS6A02G172400	P-loop containing nucleoside triphosphate hydrolase, aminotransferase	-

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
