# Peer review of "Mapping QTL for Phenological and Grain-Related Traits in a Mapping Population Derived from High-Zinc-Biofortified Wheat"

_plants, 2023, doi:10.3390/plants12010220_

Round 1

Reviewer 1 Report

This manuscript did the mapping QTL for phenological and grain-related traits including heading (DH), days to maturity (DM), plant height (PH), thousand kernel weight (TKW), and test weight (TW), from a mapping population  of wheat. The results are plentiful and the information are useful. After reviewing, I suggest this manuscript could be accepted after revisions. The suggestions or the comments are as below:

1: the title needs to be more precise written, and use Zinc

2: Line 91-97 and Line 100-105, there are so many references, please try to group them with different points

3: give some introductions for the RIL population, how did them generated, what the main features, and why the authors used them?

4: try to use the sub-titles in the Discussion part, so that it would be more clearly presented.

5: I think the next steps of the validating work is the key for almost all the QTL mapping research. So I suggest the authors could give more prospectives in the Discussion and Conclusion parts.

Author Response

Please find enclosed detailed response to reviewer comments

Reviewer 2 Report

This is a very good job,Genomic regions governing days to heading (DH), days to maturity (DM), plant height 9 (PH), thousand kernel weight (TKW), and test weight (TW) were investigated in a set of 190 RILs, which would be helpful to the future work (Character analysis, assisted breeding and so on). However, there are some areas that need improvement, such as Citation format of literature, Good quality pictures, Why do "DH, DW, PH" have 3 years of data, but "TW, TKW" only have 2 years of data, The discussion part is a bit of a repetition of the result content of 3, and so on. Besides, the DArTseq markers were not introuduced, how does the marking involve and other information of the marking can be reflected in the attachment.

Author Response

(The authors gave the same response as above.)

Reviewer 3 Report

Dear Author, 

Thank you for your work. Days to heading (DH), days to maturity (DM), plant height 9 (PH), thousand kernel weight (TKW), and test weight (TW) are significant agronomy factors in wheat. The authors invested a large time to create a RILs population and identify the QTLs over three years and in different environments. It was excellent that you have identified 37 novel QTLs related to these agronomy traits and have analyzed the potential candidate genes. 

I suggest accepting the manuscript after minor revision.

For your manuscript, I have a little few comments.

1. Line#14-15, you may keep a consistent number type (Arabic pattern).

2. Since figure 3 is one of the significant figures in your manuscript, I request a higher resolution figure.

3. I suggest making table 4 wider similar to table 3.  

Author Response

(The authors gave the same response as above.)

Reviewer 4 Report

The manuscript presents the results of mapping of agronomically important traits in mapping population obtained with the participation of a variety enriched with Zn.

Major comments:

The Introduction and Discussion chapters provide information on the application of GWAS and classical genetic mapping to localize loci that control a phenotypic manifestation of agronomically important traits (yield and its components, plant height, phenological phases, etc.). However, the purpose of the study does not follow from the information provided.

At present, a sufficient number of data have been published in which a lot of QTL for phenological phases, yield and productivity traits has been mapped and new loci candidate genes have been identified. The authors provide 31 references for GWAS and 21 references for biparental mapping. That is, there is a sufficient amount of data on the localization of QTL for the TKW, plant height, heading time, ripening time and so on. The genetic control for these traits has been studied in detail using a large number of models and mapping populations, gene-candidates were postulated. Therefore, from the Introduction section, it is not entirely clear why it is necessary to conduct QTL mapping for this particular population Kachu x Zinc-Shakti. The choice of the traits for QTL mapping is also unclear. 

The authors claim (lines 115-118) “Thus, more systematic efforts may be required to dissect the genetic mechanisms of yield and component traits in wheat and to devise marker-aided breeding approaches that involve the marker-assisted selection or genome-wide selection to obtain rapid genetic gains”

What does "systematic" mean? It is necessary to map loci in many-many different mapping populations and constantly accumulate data? 

The authors should reсonsider the data obtained and reformulate the purpose of the study. In this version of the manuscript there is no novelty and relevance of the study.

The Discussion is large, but not informative. I encourage authors to compare their data with literature data. For example, with data on localization of the known genes Vrn, Ppd, Rht, etc., in order to understand the novelty of the study.

Section Materials and Methods. Please briefly describe the methods of genotyping and mapping, and not just a reference to previously published works. Add methods for statistical processing of the results. Figure 2 shows that the distribution of the traits is not normal. Indicate which statistical test was used to evaluate correlations between traits. Add data on meteorological conditions for the years of the experiment.

Section Results. Please provide broad sense heritability for each trait and ANOVA data. 

Minor remarks:

1) Probably some words appear to be missing in the title of the article.

2) Two sentences duplicate each other (lines 14-17).

3) Too many citations (31 citations) to show the use of GWAS to detect crop QTL and associated traits (line 91-98). 

4) The same for biparental mapping populations (21 refs), lines 100-105

5) It is necessary to unify the abbreviations for traits generally accepted in the literature. For example, heading date usually referred to as HD (Heading date) and not DH

6) The quality of figures 1 and 2 is low, the symbols on the axes are not visible

7) Table 1. Please indicate the environments for which the correlations between traits were calculated.

8) Table 3. Make normal formatting, in present forms the data looks incorrectly.

9) Figure 3. The same loci have the same color, but are located in different places on the chromosomes (for example, QPh-3D). If this is the positions the same loci in different years, then appropriate indication should be made. I did not find the QPH-2B.1 locus in Table 3, while this locus is present in Figure 3.

10) Table 4. No designation of the data presented in the columns. What do the letters A and B mean in the column Marker Type? What does the RIL number mean? What does the marker names in Markers column means (Flanking markers? Or something else?). 

11) The list of references is presented in a chaotic manner. 

Author Response

(The authors gave the same response as above.)

Round 2

Reviewer 4 Report

The authors took into account most of my comments. However, the Introduction did not show the relevance and novelty of the study. Please,try again.

Major remarks:

1) You consider that there are actually no enough study that use a large number of RILs, a large number of markers and there are no rich chromosome maps that allow more accurate mapping of loci (lines 127-132).

This is not entirely true. There are a sufficient number of articles on this topic. Two examples below.

- Li  et al. (2018) Genome-wide linkage mapping of yield-related traits in three Chinese bread wheat populations using high-density SNP markers. Theor Appl Genet 131:1903–1924 903–1924 https://doi.org/10.1007/s00122-018-3122-6

Authors used three RIL populations (275 RILs, 176 RILs and 273 RILs) for construction high-density linkage maps and QTL mapping. The populations were genotyped using the wheat 90 K iSelect SNP. Number of the SNP markers for QTL mapping after filtering was more than 10000. Linkage groups looks more saturated than chromosome maps in your study.

- Ren T. (2021). Utilization of a Wheat55K SNP array-derived high-density genetic map for high-resolution mapping of quantitative trait loci for important kernel-related traits in common wheat. Theor. Appl. Genet. 134, 807–821. doi: 10.1007/s00122-020-03732-8

Authors used 371 RILs and Wheat55K SNP array (genetic map consists of 11,583).

 2) You consider that “very few studies attempted to capture the QTLs governing phenological and grain-related traits in biofortified genetic background».

This is not entirely true. Please, look at publication:

Shariatipour N et al (2021) Comparative Genomic Analysis of Quantitative Trait Loci Associated With Micronutrient Contents, Grain Quality, and Agronomic Traits in Wheat (Triticum aestivum L.). Front. Plant Sci. 12:709817. doi: 10.3389/fpls.2021.709817

The authors “conducted a meta-analysis to identify the most stable QTLs for grain yield (GY), grain quality traits, and micronutrient contents in wheat. A total of 735 QTLs retrieved from 27 independent mapping populations reported in the last 13 years were used for the meta-analysis”.

Minor remarks:

I recommend to change text between lines 91-122 in the next edition:

a)           Lines 91-93. In previous studies, GWAS panels have been phenotyped in a range of production conditions including drought, irrigated, heat, and salt stress to identify QTLs associated with grain yield and its contributing traits through GWAS (5-6 references from lines 93-102)

b)           lines 113-115 Similarly, several QTLs 113 associated with yield and contributing traits have been identified through bi-parental populations-based QTL mapping (5-6 references from lines 116-122).

The rest of the text can be deleted.

The quality of Figure 1 is very low

Author Response

Dear Editor,

Please find attached response to reviewer 4 comments.